# Impact of SARS-CoV-2 pandemic among health care workers in a secondary teaching hospital in Spain

Javier Garralda Fernandez[1], Ignacio Molero Vilches[1], Alfredo Bermejo Rodríguez[1,2‡], Isabel Cano Torres[1], Elda Isabel Colino Romay[3], Isabel García Arata[1], Jerónimo Jaqueti Aroca[1], Rosa Lillo Rodríguez[1], Daniel López Lacomba[1], Luis Mazón Cuadrado[3], Laura Molina Esteban[1], Luis Javier Morales García[1], Laura Moratilla Monzo[2,4‡], Elva Nieto-Borrajo[1], María Pacheco Delgado[1], Santiago Prieto Menchero[1], Cristina Sánchez Hernández[1], Eva Sánchez Testillano[3], Jesús García-Martínez[1,5‡]*

1 Laboratory Medicine, Hospital Universitario de Fuenlabrada, Fuenlabrada, Spain, 2 Health Sciences Faculty, Universidad Rey Juan Carlos, Madrid, Spain, 3 Occupational Health Service, Hospital Universitario de Fuenlabrada, Fuenlabrada, Spain, 4 Preventive Medicine Service, Hospital Universitario de Fuenlabrada, Fuenlabrada, Spain, 5 Escuela Internacional de Doctorado, Universidad Rey Juan Carlos, Madrid, Spain

☯ These authors contributed equally to this work.
‡ These authors also contributed equally to this work.
* jgmartinez@salud.madrid.org

**Data Availability Statement:** All relevant data are within the paper.

## Abstract

### Background

The COVID-19 pandemic has posed a huge challenge to healthcare systems and their personnel worldwide. The study of the impact of SARS-CoV-2 infection among healthcare workers (HCW), through prevalence studies, will let us know viral expansion, individuals at most risk and the most exposed areas in healthcare organizations. The aim of this study is to gauge the impact of SARS-CoV-2 pandemic in our hospital workforce and identify groups and areas at increased risk.

### Methods and findings

This is a cross-sectional and incidence study carried out on healthcare workers based on molecular and serological diagnosis of SARS-CoV-2 infection. Of the 3013 HCW invited to participate, 2439 (80.9%) were recruited, including 674 (22.4%) who had previously consulted at the Occupational Health Service (OHS) for confirmed exposure and/or presenting symptoms suggestive of COVID-19. A total of 411 (16.9%) and 264 (10.8%) healthcare workers were SARS-CoV-2 IgG and rRT-PCR positive, respectively. The cumulative prevalence considering all studies (IgG positive HCW and/or rRT-PCR positive detection) was 485 (19.9%). SARS-CoV-2 IgG-positive patients in whom the virus was not detected were 221 (9.1%); up to 151 of them (68.3%) did not report any compatible symptoms nor consult at the OHS for this reason. Men became more infected than women (25% vs 18.5%, p = 0.0009), including when data were also classified by age. COVID-19 cumulative prevalence among the HCW assigned to medical departments was higher (25.2%) than others, as well as among medical staff (25.4%) compared with other professional categories (p<0.01).

**Funding:** The author(s) received no specific funding for this work.

**Competing interests:** The authors have declared that no competing interests exist.

## Conclusions

The global impact of the COVID-19 pandemic on HCW of our centre has been 19.9%. Doctors and medical services personnel have had the highest prevalence of SARS-CoV-2 infection, but many of them have not presented compatible symptoms. This emphasizes the performance of continuous surveillance methods of the most exposed health personnel and not only based on the appearance of symptoms.

## Introduction

At the end of 2019 a cluster of patients with a severe respiratory syndrome, whose cause was finally identified as a new coronavirus (SARS-CoV-2) [1], emerged in the Chinese city of Wuhan. In January 2020, World Health Organization issued a statement for an international emergency outbreak and on March 11, 2020 declared COVID-19 a pandemic [2].

This new virus is mainly spread by secretions and respiratory droplets, thus close contact is the main way of infection, although it could also occur by other means such as fomites and contaminated surfaces [3]. Asymptomatic carriers may play an important role and contribute to the disease spread too [4]. The mean incubation period is 5.1 days (95% CI, 4.5 to 5.8 days) [5] and the basic reproductive number was estimated to be between 2.8 and 3.3 [6].

Healthcare workers (HCW) are at high risk of infection and a source of transmission for patients and other staff. In China, more than 2000 cases among health personnel were infected as of February, most in Hubei province [7]. In March 2020, Italy reported 2600 infections and more than 40 deaths among HCW population [8]. In Spain, since the beginning of the pandemic up to May 11, 40921 cases were reported to the national epidemiological surveillance network [9]; they were mainly women (76.5%) and had an average age of 46 years (36–55). Most of them presented cough (69.6%), fever (65.9%) and sore throat (39.6%). Four thousand one hundred seventy seven (10.5%) required hospitalization and 310 (1.1%) intensive care; up to 53 HCW have died. These events have resulted in lack of health personnel and, therefore, an additional burden in the fight against the pandemic.

Healthcare personnel are a key element in managing this Covid-19 pandemic. Seroprevalence studies allow estimating the individual and areas at greatest risk. This information is relevant to implement approaches to promote HCW safety and adequately manage resources in future outbreaks.

The aim of this study is to gauge the impact of SARS-CoV-2 pandemic in our hospital workforce and identify groups and areas at increased risk.

## Materials and methods

### Setting

This is a cross-sectional and incidence study carried out with HCW from the Hospital Universitario de Fuenlabrada. The hospital is located in the municipality of Fuenlabrada, south of the metropolitan area of Madrid (Spain), and it has assigned a population of about 225,000 inhabitants from the towns of Fuenlabrada, Moraleja de Enmedio and Humanes [10].

### Design and analytical methods

The cross-sectional study was carried out between April 14 and May 13, 2020 and included all HCW who came from different hospital services and belonged to all professional categories

(administrative and auxiliary services staff, central services technicians, cleaning staff, clinic assistants, doctors, nurses and watchmen). All HCW were invited to participate, recruited from hospital Human Resources database (as of April 10, 2020) by the Occupational Health Service (OHS) and summoned by the Admission Service coordinated with the hospital's Laboratory Medicine to perform the tests. A nasopharyngeal swab and venous blood sample were obtained simultaneously from all participants for molecular and serological diagnosis of SARS-CoV-2 infection, respectively. Both samples were sent during the next hour after collection and processed at the Medicine Laboratory. Nasopharyngeal samples were collected with flocked swabs in a viral transport medium that contains guanidine salts to inactivate and preserve the virus (Mole Bioscience, Taizhou, China). Nucleic acid extraction was performed in the QIAsymphony SP instrument with the QIAsymphony DSP Virus/Pathogen Midi Kit (Qiagen, Hylden, Germany) from 400 μl of sample or manually using the High Pure RNA Isolation Kit (Roche Diagnostics GmbH, Mannheim, Germany) from 200 μl of sample. Molecular detection was carried out by rRT-PCR in a LightCycler 480 System (Roche Diagnostics GmbH, Mannheim, Germany) using the LightMix® Modular SARS-CoV (COVID19) kit (Roche Diagnostics GmbH, Mannheim, Germany). Positive and negative controls as well as an internal control (LightMix® Modular EAV RNA Extraction Control) were included in each run. Serum IgG and IgM antibody directed against SARS-CoV-S (spike) and SARS-CoV-N (nucleocapside) recombinant antigens were measured in the Maglumi 2000 platform (Snibe diagnostic, Shenzhen, China) with the Maglumi 2019-nCoV (SARS-CoV-2) IgM and IgG kits in a fully automated chemiluminescence immunoassay (CLIA). The results were expressed in AU/mL and considered positive or negative following manufacturer's instructions. Once the rRT-PCR and the immunological study were analysed, a COVID status assessment report was prepared for each HCW, in which the clinical situation and symptom onset dates (if any) were assessed together with the test results carried out. A HCW was classified as asymptomatic if genetic material from SARS-COV-2 and/or serum IgG anti SARS-CoV-2 was detected but did not consult at the OHS due to compatible symptoms with COVID-19 infection [11].

The incidence study was carried out from the appearance of the first confirmed COVID-19 case in the hospital March 2, 2020 until May 13, and included those HCW who consulted at the OHS for confirmed exposure and/or presenting symptoms suggestive of COVID19 [11], as registered in the OHS database. For molecular detection of SARS-CoV-2 infection, at least one nasopharyngeal swab was obtained in viral transport medium and processed as stated previously. All these symptomatic workers were also included in the cross-sectional study as participants.

The following variables were collected from the participants in both studies: age, gender, assigned service, professional category and symptom onset dates (if any), as registered in the OHS and Human Resources Service database and HCW medical records.

## Statistical analysis

Absolute and relative prevalences were described both for SARS-CoV-2 rRT-PCR, IgG and IgM positive and negative cases and their combinations in reference to the total population recruited in both studies. Cumulative prevalence (IgG positive and/or rRT-PCR positive in any of both studies, as a proportion of the total participant population) was also described grouping by sex, age, hospital service and professional category. We have tested associations between categorical variables using $\chi 2$ test or Fischer's exact test and between continuous variables with T-Student test. A p-value $< 0.05$ was considered significant. The odds ratio was calculated to compare the prevalence rate between professional categories.

## Ethical considerations

The study has been independently evaluated and approved by the Research Ethics Committee of the Hospital Universitario de Fuenlabrada (Internal Code 20/37). An informed consent was not obtained since its need was waived by the Ethics Committee.

# Results

## Cross-sectional study

Of the 3013 HCW invited to participate in the cross-sectional study, 2439 (80.9%) were recruited, corresponding to 1911 females (78.4%) and 528 males (21.6%), with a mean age of 42.1 years (18–65).

A total of 411 HCW (16.9%) were SARS-CoV-2 IgG positive and in 32 (1.3%) IgM was detected. Only 2 out of the latter were confirmed as active SARS-CoV-2 infection by rRT-PCR, both with detectable IgG. In 11 cases, IgG was not detected and the repetitions of the tests did not confirm the infection, so they were considered IgM false positive results; in 19 cases with detectable IgG, a past infection with residual persistence of IgM antibodies was considered. So henceforth, IgM tests were not considered to determine the impact of the infection in the study population. In this study a total of 19 out of 2439 (0.8%) HCW were rRT-PCR positive, 10 out of them had an IgG antibody detectable simultaneously.

## Incidence study

Six hundred and seventy four HCW (22.4%) were also included in the incidence study because they had previously consulted at the OHS for confirmed exposure and/or presenting symptoms suggestive of COVID-19; they were 550 females (81.6%) and 124 males (18.4%), with a mean age of 42.4 years (18–65). Among this group, 245 active infections (36.4%) were reported by rRT-PCR. They declared symptoms onset dates mostly from March 1 to mid-April, although there have been cases from the last days of February to the second week of May (Fig 1). Among the 429 HCW with negative rRT-PCR, only 11 (2.6%) showed a positive IgG in the seroprevalence study.

## Global impact

In summary, 264 workers out of 2439 (10.8%) with active SARS-CoV-2 infection were detected by rRT-PCR. The combination of both studies has resulted in the global impact of the epidemic among hospital HCW. As summarized in Table 1, in 411 out of 2439 (16.9%) workers a serum IgG was recovered, while in 264 (10.8%) rRT-PCR was detected positive. In 190 (7.8%) HCW IgG and viral RNA were simultaneously detected. Most people with a negative rRT-PCR were asymptomatic, including HCW who presented detectable IgG, while those in who viral RNA was detected were mostly symptomatic. It is noteworthy that there were 221 (9.1%) SARS-CoV-2 IgG-positive patients in whom the virus was not detected by rRT-PCR; up to 151 of them (68.3%) did not report any compatible symptoms nor consult at the OHS for this reason. Up to 74 workers (3%) with confirmed SARS-CoV-2 infection by rRT-PCR had not achieved IgG seroconversion at the time the study finished. In brief, the overall cumulative prevalence considering all studies (IgG positive HCW and rRT-PCR positive detection without seroconversion) has been 485 infected HCW out of 2439 (19.9%).

Significant differences have been detected when the prevalence data have been stratified by sex. As shown in Table 2, men became more infected than women (25% vs 18.5%, p = 0.0009). When data were also classified by age, again the COVID19 epidemic has infected more men, particularly in the age group between 35 and 45 years (32.1% vs 17.7%) (Fig 2), although these differences were not significant (p = 0.38).

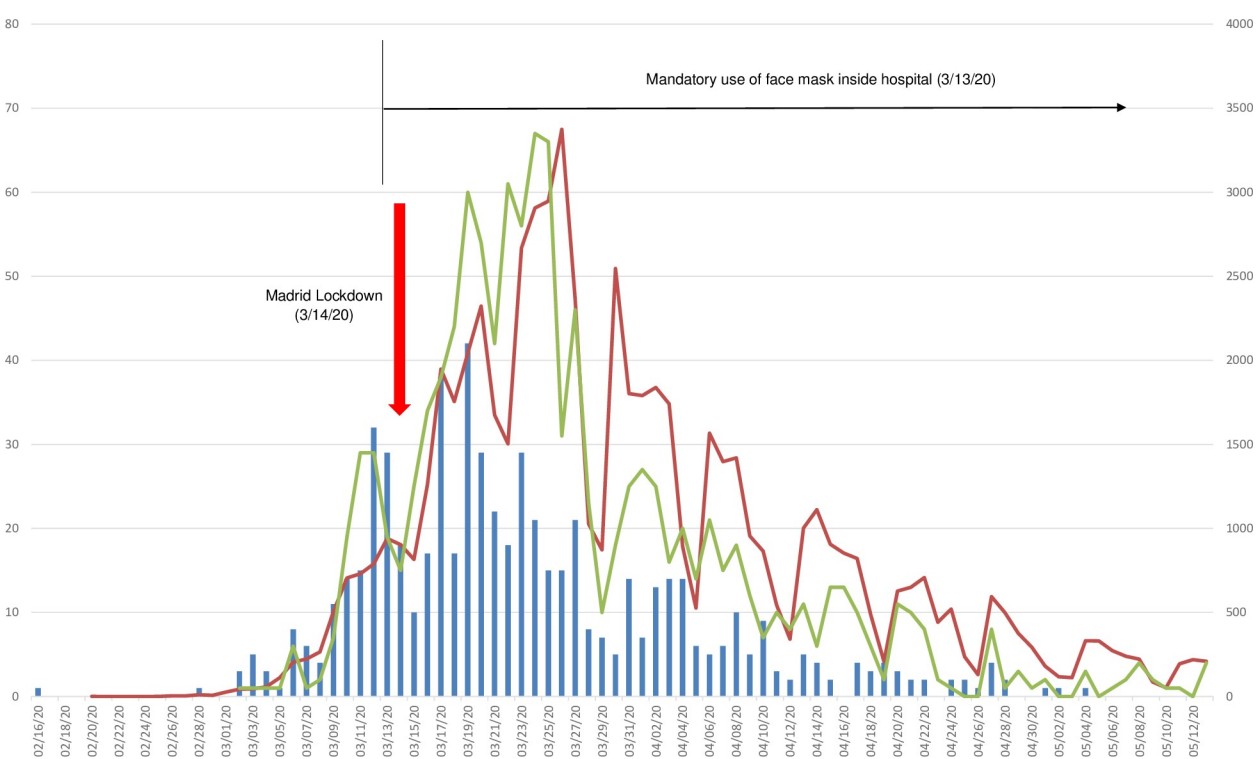

**Fig 1. Epidemiologic curve from symptoms onset dates in relation to cases diagnosed in Madrid.** Blue columns are the number of HCW from HUF that declared COVID-19 compatible symptoms by date of symptom onset; red line is rRT-PCR-based diagnosed cases by date of diagnosis in Community of Madrid*; green line is rRT-PCR-based diagnosed cases by date in HUF **. HCW: healthcare workers; HUF: Hospital Universitario de Fuenlabrada. *Source: based on RENAVE data [12]. ** Source: based on own Laboratory Medicine data.

In order to study the influence of work-related exposure to the virus into the possibility of getting infected, HCW were classified according to their medical department. Cumulative prevalence of SARS-CoV-2 among the HCW assigned to the Internal Medicine-Emergency (ER) (29.8%) departments was higher than other departments (Fig 3). Personnel assigned to auxiliary and administrative services were those with the lowest risk of infection (13.7%). Difference between groups were statistically significant (p<0.05).

Finally, we analysed the influence of the staff category and the prevalence of SARS-CoV-2 in our hospital. A shown in Fig 4, cumulative prevalence of SARS-CoV-2 was higher among the medical staff (25.4%) compared with other categories (p<0.01). The odds of being infected by staff category was 2.13 (1.53–2.96, p<0.0001), 1.77 (1.26–2.49, p = 0.0009), 1.68 (1.02–2.78,

**Table 1. Prevalence of symptomatic and asymptomatic cases based on SARS-CoV-2 RT-PCR and IgG status.**

| | IgG+ n (%) | | IgG- n (%) | | Total n (%) |
|---|---|---|---|---|---|
| | symptomatic | asymptomatic | symptomatic | asymptomatic | |
| **rRT-PCR+** | 178 (93.7) | 12 (6.3) | 66 (89.2) | 8 (10.8) | 264 (10.8) |
| | 190 (7.8) | | 74 (3.0) | | |
| **rRT-PCR-** | 70 (31.7) | 151 (68,3) | 287 (14.7) | 1667 (85.3) | 2175 (89.2) |
| | 221 (9.1) | | 1954 (80.1) | | |
| **Total** | 248 (60.3) | 163 (38.7) | 353 (17.4) | 1675 (82.6) | 2439 |
| | 411 (16.9) | | 2028 (83.1) | | |

**Table 2. Frequency of SARS-Cov-2 rRT-PCR and IgG result cases by sex.**

|  | Women | Men | Total |
|---|---|---|---|
|  | n = 1911 (%) | n = 528 (%) | n = 2439 (%) |
| PCR positive and IgG positive | 142 (7.5%) | 48 (9.1%) | 190 (7.8%) |
| PCR positive and IgG negative | 56 (2.9%) | 18 (3.4%) | 74 (3.0%) |
| PCR negative and IgG positive | 155 (8.1%) | 66 (12.5%) | 221 (9.1%) |
| PCR negative and IgG negative | 1558 (81.5%) | 396 (75.0%) | 1954 (80.1%) |
| IgG positive and/or PCR positive | 353 (18.5%) | 132 (25.0%) | 485 (19.9%) |

Sex differences in cumulative prevalence (IgG positive and/or RT-PCR positive) were significant (p = 0.0009).

p = 0.0435), 1.51 (1.09–2.09, p = 0.0129), 1.39 (0.87–2.21, p = 0.1685), and 1.27 (0.68–2.39, p = 0.4541) for doctors, nurses assistants, watchmen, nurses, technicians, and cleaning staff, respectively.

## Discussion

The main objective of this work was to describe the global impact of the COVID-19 epidemic among the workers of a second level hospital located in an urban population in one of the areas with the highest incidence in Spain [13]. There have been two data sources: a cross-sectional study of seroprevalence by measuring the proportion of IgG and IgM antibodies against SARS-CoV-2 in all hospital personnel and a longitudinal study of the incidence of infection among those HCW who had consulted for symptoms compatible with COVID-19 infection and/or unsafe contact or exposure to a confirmed case. As it has been shown in the results, the combination of both data sources has allowed us to have a more complete picture of the pandemic impact in our center, since there have been up to 74 patients (3% of the total HCW)

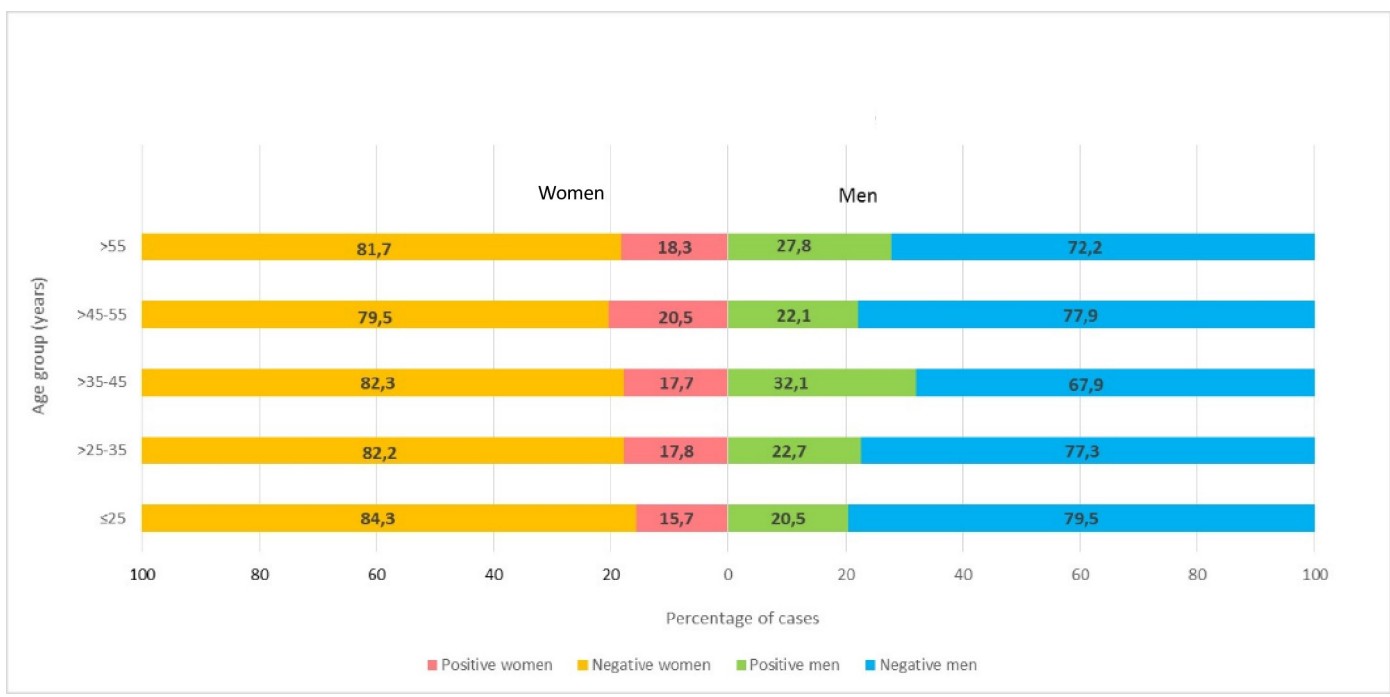

**Fig 2. Age and sex cumulative prevalence of SARS-CoV-2 cases (IgG positive and/or RT-PCR positive).** The differences by age group were not significant.

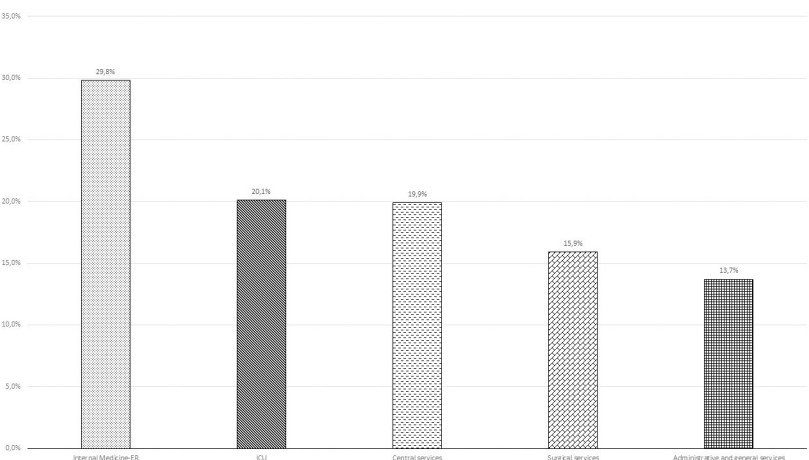

**Fig 3. Cumulative prevalence of SARS-CoV-2 cases (IgG positive and/or RT-PCR positive) by hospital departments.** Differences were significant (p<0.001).

who have had a confirmed infection by a positive rRT-PCR but have not seroconverted; and otherwise there were 151 workers (6.2%) in whom the virus had not been detected nor had compatible symptoms, but an IgG antibody has been detected.

The cumulative prevalence of SARS-CoV-2 infection in our center has been of 19.9%. This proportion is significantly higher than that of the population in the area in which the center is

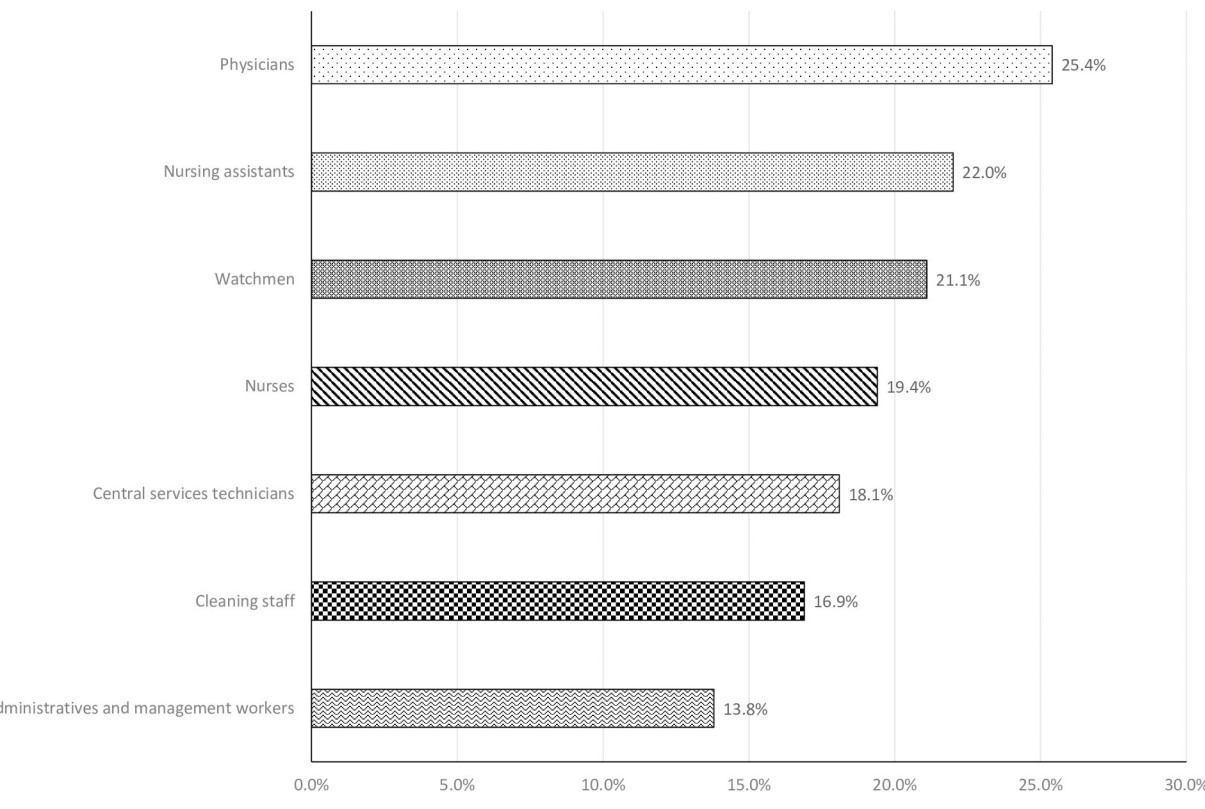

**Fig 4. Cumulative prevalence of SARS-CoV-2 cases (IgG positive and/or RT-PCR positive) by professional category.** Differences were significant (p<0.05).

located (11.4%), as stated in the wide sero-epidemiological study ENE-COVID19 [13], despite the fact that symptomatic cases have followed a very similar onset kinetics (Fig 1). It is very likely that there has been a common source of contagion with that of the general population, but also it is clear that there has also been a risk exposure, especially in the first days of the outbreak when the virus situation was not exactly known and HCW were not taking the appropriate protective measures [14].

There have been several recent publications that have reported different data of infection that affects healthcare personnel. Korth et al have published an estimated seroprevalence in a German center of only 1.6% [15]. A hospital-wide survey in Belgium rendered a 6.4% of seroprevalence [16]. In another Spanish reference center in Barcelona, a cumulative prevalence has been calculated in 578 hospital workers through the detection of IgG, IgM and IgA and viral RNA of 11.6%, very similar to the population in which it is found [17]. On the contrary, in other sero-epidemiological study in a nearby hospital center, a seroprevalence of 31.6% has been reported, a much higher level of infection than expected [18]. Although the population assigned to this center has a cumulative incidence higher than that of our center, 1,133.63 vs. 845.12 cases/100,000 inhabitants (a 26% higher incidence rate) [19], it exceeds the prevalence we have found in our survey by 37% (31.6 vs 19.9%). Additional data and studies are required to justify this difference.

Among the data from our study, it is noteworthy the high number of HCW who have been infected and who have not declared any COVID-19 compatible symptoms. One hundred and sixty-three of the 411 (38.7%) workers who presented IgG were asymptomatic. Even in 12 of them (3%), SARS-CoV-2 was detected by rt-PCR.". The role of asymptomatic patients has been described both in general population surveys (40–45%) [20] and in other studies among healthcare personnel, but with highly variable prevalence data ranging from 0.74 to 48.5% [18, 21–24]. Transmission of the virus from the asymptomatic cases has already been demonstrated [4, 25] and the free movement of HCW is a risk for new infections, both to other health personnel, patients and even the community (relatives, friends, etc.). Therefore the need for continued surveillance is critical, at least while community transmission of the virus continues. Seroepidemiological studies are a useful tool in the identification of these asymptomatic patients and in understanding the prevalence of COVID-19 [26, 27].

Participants in our survey were mainly female (78.4%). But when we stratified data by sex, significant differences have been detected and more men have been in contact with virus. In the 2002 SARS outbreak, it was already described that men were more severely affected than women [28]. This same characteristic is occurring in the COVID-19 pandemic. Men have a higher risk of being seriously ill and there has even been an excess mortality associated with the male sex [29, 30]. Among HCW at our centre, men have been infected globally more than women, although we have not registered any sign of severity. This fact has also been described in other centres [18, 21], but this is the first report in which the proportion of SARS infections is significantly different depending on sex. The cause could be related to the protection that women have by immunity mechanisms linked to the X-chromosome, different levels of sex hormones and levels of expression of the receptor for the angiotensin-converting enzyme 2, which is one of the entrance doors of the virus in the cell [31, 32].

Our data shows two main populations that have been especially exposed to COVID-19 infectious patients. Firstly, doctors have had a higher percentage of infections (25.4%) than the rest of the professional categories. Secondly, the workers who have carried out their task in the medical and emergency services have presented a higher prevalence (29.8%) of infection than the rest of the staff in other services. Other studies in our environment have not described these differences [17, 18]. On the contrary, Chen et al. [21] have shown that while exposed to COVID-19 patients, doctors might have higher risk of seroconversion, compared with HCW

exposed to colleagues, but this difference was also observed between nurses and general services assistants. It is evident that among HCW with higher prevalence there has been a greater exposure to infectious patients [18]. It is probably that personal protection measures could not have been used properly in the first days of the epidemic but personal protection equipment shortages also occurred, which is a direct cause of HCW infection [33].

Although our study has shown an association between being a HCW and becoming COVID-19 positive, it has not demonstrated a cause and effect. However that objective was beyond the scope of the study. The appearance of positive cases between the general population and HCWs has in fact been almost parallel in terms of evolution (Fig 1) but very different in terms of prevalence. HCWs are part of the general population and participate in social and family life in a similar way to the rest, so their evolution is highly influenced. Nevertheless, they have also been exposed to positive cases due to their job, which could explain their higher prevalence than in the general population. As the positive patients increased, so did the exposure of the workers and vice versa.

In the temporal evolution of confirmed cases registry among health personnel (Fig 1), the decrease in cases that began 7–14 days after the imposition of measures to contain the spread of the virus is striking. Actions included maintaining social distance, hand washing and hygiene, and especially mandatory use of a facial mask. The main health authorities, such as WHO [34], recommended these arrangements. Although the control of the viral expansion cannot be attributed in an absolute way to these measures, since the Madrid lockdown coincided in time, it is undoubted that this basic measure is one of the actions that have allowed the control of infection in a highly exposed population as HCW.

Our study presents some pitfalls that deserve further comments. First, although the participation rate has been 80.9%, a selection bias could have been introduced. Nevertheless, this participation rate is higher than that reported in other similar studies [16, 18, 35]. In addition, all professional categories and age ranges have been represented. Secondly, a professional who has fallen ill and may have consulted at other health centre or primary care facility and not at the OHS, would have introduced a new bias. Instead, we could be aware of this information since the sick leave was processed at our hospital Occupational Health department. Also, some doubts have been raised about the performance of the serological and molecular diagnostic tools employed, since there have been 74 workers who have been infected but have not seroconverted and 11 people who have presented symptoms and have been seroconverted but have not been diagnosed by rRT-PCR. These results could have underestimated the real impact of the pandemic in our study population. Although we have not evaluated the sensitivity and specificity of the diagnostic tools used in our study, there is evidence on the performance of the different platforms and targets. In the case of rt-PCR, sensitivity ranges of 43.6–100% and specificity of 98–100% depending on protocols and targets [36]. These data are also influenced by the sample type and quality and the time since exposition [37]. To resolve the risk of high false negative rate, we have established measures such as the repetition of molecular tests in the case of symptomatic HCW or confirmed case contacts that had been negative. We have also standardize sampling and reduced the sample extraction team reinforcing its training. On the other hand, serum IgG antibody detection have been validated by other researchers with excellent results, achieving 100% sensitivity at day 12 post infection [38, 39]. Test performance could also be affected by the prevalence of infection in the study population. However, as Lorentzen et al. [40] have verified, with a prevalence> 10%, as seen in our center, the positive and negative predictive values remain high even with sensitivity values <80%.

Finally, our study design has introduced some limitations. Cross-sectional studies do not allow causal inferences and only association and no causation can be inferred. There may also be an overestimation of results as well as temporal bias, such as the appearance of new cases in

a continuously evolving epidemic. With this design we are unable to investigate temporary relation between exposures and outcomes. Also incidence studies do not allow causal inferences since they lack a control group. In addition, there is the possibility of loss of individuals during follow-up.

## Conclusions

In conclusion, the global impact of the COVID-19 pandemic on HCW of our centre has been 19.9%. This prevalence has been significantly higher than general population. Similarly, doctors and medical services personnel have had the highest prevalence of SARS-CoV-2 infection, but many of them have not presented compatible symptoms. This emphasizes the importance of the performance of continuous surveillance methods of the most exposed health personnel and not only based on the appearance of symptoms. These methods should include both antibody and viral detection methods to have a more realistic picture of the virus circulation in a certain population.

## Acknowledgments

Our deep acknowledgment to all the staff of the Hospital Universitario de Fuenlabrada for their tireless fight against the pandemic. Thanks to all the volunteers who have participated in this study. We would also like to appreciate the staff of the laboratory for their excellent technical assistance.

## Author Contributions

**Conceptualization:** Daniel López Lacomba, Santiago Prieto Menchero, Jesús García-Martínez.

**Data curation:** Alfredo Bermejo Rodríguez, Jerónimo Jaqueti Aroca, Rosa Lillo Rodríguez, Daniel López Lacomba.

**Formal analysis:** Javier Garralda Fernandez, Alfredo Bermejo Rodríguez, Jerónimo Jaqueti Aroca, Rosa Lillo Rodríguez, Daniel López Lacomba, Laura Moratilla Monzo, Elva Nieto-Borrajo, María Pacheco Delgado, Santiago Prieto Menchero, Jesús García-Martínez.

**Investigation:** Elda Isabel Colino Romay, Isabel García Arata, Jerónimo Jaqueti Aroca, Rosa Lillo Rodríguez, Luis Mazón Cuadrado, Laura Molina Esteban, Luis Javier Morales García, Cristina Sánchez Hernández, Eva Sánchez Testillano.

**Methodology:** Ignacio Molero Vilches, Luis Mazón Cuadrado, Elva Nieto-Borrajo, María Pacheco Delgado, Cristina Sánchez Hernández, Jesús García-Martínez.

**Software:** Daniel López Lacomba.

**Supervision:** Jesús García-Martínez.

**Validation:** Alfredo Bermejo Rodríguez, Laura Moratilla Monzo.

**Writing – original draft:** Javier Garralda Fernandez, Ignacio Molero Vilches, Alfredo Bermejo Rodríguez, Isabel Cano Torres, Isabel García Arata, Jerónimo Jaqueti Aroca, Rosa Lillo Rodríguez, Laura Molina Esteban, Cristina Sánchez Hernández.

**Writing – review & editing:** Luis Javier Morales García, Laura Moratilla Monzo, Santiago Prieto Menchero, Jesús García-Martínez.

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
