## [Decision Letter · Decision Letter 0]

28 Oct 2020

PONE-D-20-22318

Impact of SARS-CoV-2 pandemic among health care workers in a secondary teaching hospital in Spain.

PLOS ONE

Dear Dr. García-Martínez,

Thank you for submitting your manuscript to PLOS ONE. After careful consideration, we feel that it has merit but does not fully meet PLOS ONE’s publication criteria as it currently stands. Therefore, we invite you to submit a revised version of the manuscript that addresses the points raised during the review process.

We look forward to receiving your revised manuscript.

Kind regards,

Kimberly Page, PhD, MPH

Academic Editor

PLOS ONE

Journal Requirements:

2.Please provide additional details regarding participant consent. In the ethics statement in the Methods and online submission information, please ensure that you have specified (1) whether consent was informed and (2) what type you obtained (for instance, written or verbal, and if verbal, how it was documented and witnessed). If your study included minors, state whether you obtained consent from parents or guardians. If the need for consent was waived by the ethics committee, please include this information.

Additional Editor Comments (if provided):

I concur with Reviewers' comments. The paper presents interesting results, but revisions are in order .

Reviewers' comments:

Reviewer's Responses to Questions

**Comments to the Author**

1. Is the manuscript technically sound, and do the data support the conclusions?

Reviewer #1: Yes

Reviewer #2: Yes

2. Has the statistical analysis been performed appropriately and rigorously? 

Reviewer #1: Yes

Reviewer #2: Yes

3. Have the authors made all data underlying the findings in their manuscript fully available?

Reviewer #1: Yes

Reviewer #2: Yes

4. Is the manuscript presented in an intelligible fashion and written in standard English?

Reviewer #1: Yes

Reviewer #2: Yes

5. Review Comments to the Author

Reviewer #1: Study Design: This appears to be a cross-sectional study only, which looks at 1) the prevalence of COVID + results of all HCW within the institution from April to May, and 2) the prevalence of COVID + employees who have a known exposure or have presented with symptoms from March to May who have reported to OHS. Cross-sectional studies look for association in a defined time period, as described in this study, where as longitudinal studies have repeated measures on the same population over time to look at cause and effect, which this study does not do.

Additional Discussion Paragraphs:

1) A more in depth discussion on the specificity and sensitivity of both NP PCR and serology testing is needed. PCR has a high false negative rate while serology testing has both a high false positive and false negative rate. Furthermore, positive predictive value is impacted by the prevalence in the populations. All of these points should be discussed as they impact the interpretation of the results.

2) Other than in figure 1, there is no mention of the drop in the number of cases after universal mask use for HCW was implemented. This is an important observation and emphasizes the importance of the use of PPE and source control.

3) In limitations, state the limitations of cross sectional studies

This study has shown an association between being a HCW and becoming COVID +, but has not demonstrated a cause and effect. This should be better described in the discussion and conclusion. Furthermore, the are rates and trajectories between the HCW and general population are very similar as shown in Fig 1, with HCW rates falling below that of the general population after mask usage was implemented, so it is difficult to describe a cause and effect of being a HCW working in a hospital when the community spread and rates follow similar curves.

Grammatical corrections:

The study of the impact of

33 areas. *Exposed areas where? In the hospital?

37 *deleter the work 'finally'

39 the "Occupational Health Services (OHS)" for confirmed exposure and/or presenting symptoms suggestive of

43 detection) "was" 485 (19.9%). SARS-CoV-2 IgG-positive patients in whom the

52 Conclusions. "The" global impact of the COVID-19 pandemic on HCW of our centre

67 means "such" as fomites and contaminated surfaces[3]. Asymptomatic carriers may play

76 cases were reported to the national epidemiological surveillance network[9]";" they

83 Health personnel are critical and have become a weak point of the health *explain what you mean by 'become a weak point"

159 *delete the word 'finally'

300 There are 163 of 411 in total (38.7%), of which there are even 12 (3%) in which the virus was detected by rRT-PCR. *Explain this sentence better. It does not do a good job interpreting the result.

Reviewer #2: The paper is nicely written and gives an good overview of the spread of SARS-CoV-2 in a Spanish hospital.

In the result section, the use of two data sources is a bit confusing. This should be clarified.

The differences in prevalence in departments is really interesting. A further differentiation into medical departments (ICU vs. ER) would be interesting.

In my opinion, the major flaw in this research is the inclusion of antibody-positivity into the overall SARS-CoV-2 rate in the study population. In the discussion the possibility of false-positive test results in IgG and IgM should be mentioned. According to this possibility using only one antibody-assay IgG-positive participants should not be counted as SARS-CoV-2-cases as done here.

Small typos need to be corrected (example: region page 9, line 185).

6. PLOS authors have the option to publish the peer review history of their article (what does this mean?). If published, this will include your full peer review and any attached files.

Reviewer #1: No

Reviewer #2: No

---

## [Author Response · Author response to Decision Letter 0]

7 Dec 2020

Response to reviewers

PLOS ONE’s style requirements have been revised.

2. Please provide additional details regarding participant consent.

We have added additional details regarding participant consent.

Reviewer #1. Study Design: This appears to be a cross-sectional study only, which looks at 1) the prevalence of COVID + results of all HCW within the institution from April to May, and 2) the prevalence of COVID + employees who have a known exposure or have presented with symptoms from March to May who have reported to OHS. Cross-sectional studies look for association in a defined time period, as described in this study, where as longitudinal studies have repeated measures on the same population over time to look at cause and effect, which this study does not do.

We agree with the reviewer that our study is not designed to establish a causal relationship, so no repeated measurements are performed over a period, nor two study cohorts exist, and therefore it cannot be defined as a longitudinal study itself. However, we believe that it is not just a cross-sectional study only because in order to know the real impact of the pandemic in our population, a continuous follow-up of the study population was carried out (through rt-PCR test) to know the incidence in HCWs who presented symptoms compatible with the COVID-19 infection or had close contact with a confirmed case. We think that perhaps it is more appropriate to define as an incidence study what we had previously defined as a longitudinal study. The incidence study is an observational longitudinal study whose main objective is to estimate the incidence of a disease in a given population. In this way, we have made the definition changes on the manuscript.

1) A more in depth discussion on the specificity and sensitivity of both NP PCR and serology testing is needed. PCR has a high false negative rate while serology testing has both a high false positive and false negative rate. Furthermore, positive predictive value is impacted by the prevalence in the populations. All of these points should be discussed as they impact the interpretation of the results.

We have included in the text the following paragraph in response to the reviewer's requirements:

Although we have not evaluated the sensitivity and specificity of the diagnostic tools used in our study, there is evidence on the performance of the different platforms and targets. In the case of rt-PCR, sensitivity ranges of 43.6-100% and specificity of 98-100% depending on protocols and targets(35). These data are also influenced by the sample type and quality and the time since exposition(36). To resolve the risk of high false negative rate, we have established measures such as the repetition of molecular tests in the case of symptomatic HCW or confirmed case contacts that had been negative. We have also standardize sampling and reduced the sample extraction team reinforcing its training. On the other hand, serum IgG antibody detection have been validated by other researchers with excellent results, achieving 100% sensitivity at day 12 post infection(37,38). Test performance could also be affected by the prevalence of infection in the study population. However, as Lorentzen et al(39) have verified, with a prevalence> 10%, as seen in our center, the positive and negative predictive values remain high even with sensitivity values <80%.

2) Other than in figure 1, there is no mention of the drop in the number of cases after universal mask use for HCW was implemented. This is an important observation and emphasizes the importance of the use of PPE and source control.

The following text has been included in the discussion section:

In the temporal evolution of confirmed cases registry among health personnel (Fig 1), the decrease in cases that began 7-14 days after the imposition of measures to contain the spread of the virus is striking. Actions included maintaining social distance, hand washing and hygiene, and especially mandatory use of a facial mask. The main health authorities, such as WHO (Advice), recommended these arrangements. Although the control of the viral expansion cannot be attributed in an absolute way to these measures, since the Madrid lockdown coincided in time, it is undoubted that this basic measure is one of the actions that have allowed the control of infection in a highly exposed population as HCW.

3) In limitations, state the limitations of cross sectional studies.

The following limitations of study design has been included in the text:

Finally, our study design has introduced some limitations. Cross-sectional studies do not allow causal inferences and only association and no causation can be inferred. There may also be an overestimation of results as well as temporal bias, such as the appearance of new cases in a continuously evolving epidemic. With this design we are unable to investigate temporary relation between exposures and outcomes. Also incidence studies do not allow causal inferences since they lack a control group. In addition, there is the possibility of loss of individuals during follow-up.

This study has shown an association between being a HCW and becoming COVID +, but has not demonstrated a cause and effect. This should be better described in the discussion and conclusion. Furthermore, the are rates and trajectories between the HCW and general population are very similar as shown in Fig 1, with HCW rates falling below that of the general population after mask usage was implemented, so it is difficult to describe a cause and effect of being a HCW working in a hospital when the community spread and rates follow similar curves.

Indeed, we agree with the reviewer that the association shown between being hcw and becoming covid + has not been proven as cause and effect. This objective was beyond the scope of the study. In the introductory section, it is already stated that the objective of the study was to describe the impact of the pandemic among health workers. The appearance of positive cases between the general population and HCWs has in fact been almost parallel in terms of evolution but very different in terms of prevalence. HCWs are part of the general population and participate in social and family life in a similar way to the rest, so their evolution is highly influenced. But they have also been exposed to positive cases due to their work, which could explain their higher prevalence than in the general population. As the positive patients increased, so did the exposure of the workers and vice versa. Following the reviewer's indication, this will be better described in the manuscript.

Grammatical corrections:

The study of the impact of

33 areas. *Exposed areas where? In the hospital?

the most exposed areas in healthcare organizations

37 *deleter the work 'finally'

done

39 the "Occupational Health Services (OHS)" for confirmed exposure and/or presenting symptoms suggestive of

done

43 detection) "was" 485 (19.9%). SARS-CoV-2 IgG-positive patients in whom the

done

52 Conclusions. "The" global impact of the COVID-19 pandemic on HCW of our centre

done

67 means "such" as fomites and contaminated surfaces[3]. Asymptomatic carriers may play

done

76 cases were reported to the national epidemiological surveillance network[9]";" they

done

83 Health personnel are critical and have become a weak point of the health *explain what you mean by 'become a weak point"

This means that from the point of view of healthcare system, HCW had been the weak link in chain: health personnel have had a great demand for work during this pandemic due to the avalanche of infected patients; as these workers are highly exposed and have more chances of becoming infected and falling ill, they endanger, even more if possible, the sustainability of the health system. We have changed that expression for “Healthcare personnel are a key element in managing this Covid-19 pandemic”.

159 *delete the word 'finally'

done

300 There are 163 of 411 in total (38.7%), of which there are even 12 (3%) in which the virus was detected by rRT-PCR. *Explain this sentence better. It does not do a good job interpreting the result.

This sentence has been changed: “One hundred and sixty-three of the 411 (38.7%) workers who presented IgG were asymptomatic. Even in 12 of them (3%) SARS-CoV-2 was detected by rt-PCR.”

Reviewer #2: The paper is nicely written and gives an good overview of the spread of SARS-CoV-2 in a Spanish hospital.

In the result section, the use of two data sources is a bit confusing. This should be clarified.

As indicated by the reviewer, and to clarify the origin of data, several subsections have been created in the results section: cross-sectional study, incidence study and global impact.

The differences in prevalence in departments is really interesting. A further differentiation into medical departments (ICU vs. ER) would be interesting.

We agree with the reviewer that these differences are very interesting. According to his indications, we have also included the differences between the Emergency-Internal Medicine and ICU groups.

In my opinion, the major flaw in this research is the inclusion of antibody-positivity into the overall SARS-CoV-2 rate in the study population. In the discussion the possibility of false-positive test results in IgG and IgM should be mentioned. According to this possibility using only one antibody-assay IgG-positive participants should not be counted as SARS-CoV-2-cases as done here.

We agree with the reviewer that the performance of diagnostic tools may be one of the main limitations of the study. Also following the instructions of reviewer #1, the limitations and consequences of the lack of sensitivity and specificity in the diagnostic techniques used to classify the study participants, including serological techniques, have been yet introduced into the manuscript. Validation of serological techniques is beyond the scope of this study. However, the excellent results obtained with the kit used by our team by other researchers supports its use as a screening and classification method. Even taking into account the limitations already included in the manuscript, we believe that the positivity of IgG antibodies against SARS-CoV-2 is an excellent marker of previous infection. This parameter has been widely used in epidemiology and infection control as a unique marker because it is a powerful indicator of present or past contact with a microorganism.

Small typos need to be corrected (example: region page 9, line 185).

Manuscript has been revised and corrected.

---

## [Editor Report · Decision Letter 1]

21 Dec 2020

Impact of SARS-CoV-2 pandemic among health care workers in a secondary teaching hospital in Spain.

PONE-D-20-22318R1

Dear Dr. Garcia-Martinez,

We’re pleased to inform you that your manuscript has been judged scientifically suitable for publication and will be formally accepted for publication once it meets all outstanding technical requirements.

Kind regards,

Kimberly Page, PhD, MPH

Academic Editor

PLOS ONE
---

## [Editor Report · Acceptance letter]

28 Dec 2020

PONE-D-20-22318R1 

Impact of SARS-CoV-2 pandemic among health care workers in a secondary teaching hospital in Spain. 

Dear Dr. García-Martínez:

I'm pleased to inform you that your manuscript has been deemed suitable for publication in PLOS ONE. Congratulations! Your manuscript is now with our production department. 

Kind regards, 

on behalf of

Dr. Kimberly Page 

Academic Editor

PLOS ONE